# Mesenchymal Stromal Cells from Healthy and Inflamed Human Gingiva Respond Differently to *Porphyromonas gingivalis*

**DOI:** 10.3390/ijms23073510

**Published:** 2022-03-23

**Authors:** Marina Bekić, Marina Radanović, Jelena Đokić, Sergej Tomić, Mile Eraković, Dušan Radojević, Miloš Duka, Dejan Marković, Milan Marković, Bashkim Ismaili, Dejan Bokonjić, Miodrag Čolić

**Affiliations:** 1Institute for the Application of Nuclear Energy, University of Belgrade, 11060 Belgrade, Serbia; marina.bekic@inep.co.rs (M.B.); sergej.tomic@inep.co.rs (S.T.); milan.markovic@inep.co.rs (M.M.); 2Medical Faculty Foča, University of East Sarajevo, 73300 Foča, Bosnia and Herzegovina; marinamilinkovic7@gmail.com (M.R.); dbokonjic@gmail.com (D.B.); 3Institute of Molecular Genetics and Genetic Engineering, University of Belgrade, 11042 Belgrade, Serbia; jelena.djokic@imgge.bg.ac.rs (J.Đ.); d.d.radojevic@gmail.com (D.R.); 4Clinic for Stomatology, Medical Faculty of the Military Medical Academy, University of Defense, 11154 Belgrade, Serbia; erakovic.nino78@gmail.com (M.E.); milosvduka@gmail.com (M.D.); 5Faculty of Dental Medicine, University of Belgrade, 11118 Belgrade, Serbia; dejan.markovic@stomf.bg.ac.rs; 6Faculty of Dental Medicine, International Balkan University, 1000 Skopje, North Macedonia; dr_baki@yahoo.com

**Keywords:** Gingiva-Derived Mesenchymal Stromal Cells, *Porphyromonas gingivalis*, cell differentiation, gene expression, inflammation, tissue regeneration

## Abstract

Gingiva-Derived Mesenchymal Stromal Cells (GMSCs) have been shown to play an important role in periodontitis. However, how *P. gingivalis*, one of the key etiological agents of the disease, affects healthy (H)- and periodontitis (P)-GMSCs is unknown. To address this problem, we established 10 H-GMSC and 12 P-GMSC lines. No significant differences in morphology, differentiation into chondroblasts and adipocytes, expression of characteristic MSCS markers, including pericyte antigens NG2 and PDGFR, were observed between H- and P-GMSC lines. However, proliferation, cell size and osteogenic potential were higher in P-GMSCs, in contrast to their lower ability to suppress mononuclear cell proliferation. *P. gingivalis* up-regulated the mRNA expression of IL-6, IL-8, MCP-1, GRO-α, RANTES, TLR-2, HIF-1α, OPG, MMP-3, SDF-1, HGF and IP-10 in P-GMSCs, whereas only IL-6, MCP-1 and GRO-α were up-regulated in H-GMSCs. The expression of MCP-1, RANTES, IP-10 and HGF was significantly higher in P-GMSCs compared to H-GMSCs, but IDO1 was lower. No significant changes in the expression of TLR-3, TLR-4, TGF-β, LAP, IGFBP4 and TIMP-1 were observed in both types of GMSCs. In conclusion, our results suggest that P-GMSCs retain their pro-inflammatory properties in culture, exhibit lower immunosuppressive potential than their healthy counterparts, and impaired regeneration-associated gene induction in culture. All these functions are potentiated significantly by *P. gingivalis* treatment.

## 1. Introduction

Chronic periodontitis (CP) is an inflammatory disease that affects the gingiva, alveolar bone, periodontal ligament, and cementum. According to the World Health Organization, CP affects 10–15% of the adult population worldwide [1,2], and in addition to caries, periodontal disease is the main cause of tooth loss in adults [3]. CP usually begins as gingivitis characterised by redness, swelling, and bleeding of the gingival tissue, but at this stage does not affect the destruction of soft and hard supporting structures, including the attachment of teeth. As the disease progresses, the inflammatory process leads to the destruction of the periodontal ligament and alveolar bone, migration of the epithelial ligament and formation of the periodontal pocket, as the main clinical feature of CP [4].

Human subgingival plaque contains more than 500 species of bacteria, but only a small proportion of them are involved in the initiation and progression of CP [4]. Amongst periodontal bacteria, *Porphyromonas gingivalis*, a Gram-negative species, has the most important etiopathogenetic role in CP [5] and was found in 85.75% of subgingival plaque samples in CP patients [6]. *P. gingivalis* produces a number of virulence factors such as various enzymes, lipopolysaccharides (LPS), exopolysaccharides, outer membrane proteins and fimbriae components. These factors could penetrate the gingival tissue and cause tissue destruction directly or indirectly, by inducing inflammation and targeting specific host cells [4,5]. CP is recognised as a polymicrobial inflammatory disease due to not only overlapping but also synergistic effects of virulence factors produced by *P. gingivalis* and other bacteria present in the periodontal pocket [7].

Despite almost all cells within the gingival tissue being targeted by *P. gingivalis*, from an immunological point of view, initiation, maintenance and progression of CP are guided by components of innate and adaptive immunity. One of the least studied cell types in CP is the gingiva-derived mesenchymal stromal cell (GMSCs). It is well known that MSCs, also referred to as multipotent mesenchymal stem cells, are present in all organs in the body where they play key roles in tissue regeneration and tissue homeostasis [8].

MSCs possess fibroblast-like morphology, show self-renewal capability and multilineage differentiation potential. Dental tissues are very rich in different populations of MSCs. They are usually homogenous, proliferate faster than bone marrow-derived MSCs, express characteristic markers, and differentiate into different cell lines of mesenchymal origin such as osteoblasts, chondroblasts and adipocytes. GMSCs possess properties that distinguish them from the rest of the dental MSCs. Namely, GMSCs are easily isolated, do not show spontaneous differentiation even after several passages, have additional neurogenic and epithelial differentiation potential due to gingival origin from the neural crest, show powerful and unique regenerative capacity in vivo, maintain morphological, phenotypical and telomerase stability after long-term cultures [9] and have excellent immunomodulatory and anti-inflammatory properties in vitro and in vivo [4,9,10]. In addition, GMSCs also differentiate into synoviocytes and endothelial cells under specific in vitro culture conditions [9]. Much the same as bone marrow MSCs and most dental MSCs, GMSCs show strong and stable expression of MSC surface markers such as CD44, CD29, CD73, CD90 and CD105, variable expression of CD146, CD56 and Stro-1 and negative expression of hematopoietic stem cell markers (CD34, CD14, CD11b, and CD45). GMSCs also express Oct4 and Nanog embryonic stem cell markers, nestin, a neuronal stem cell marker, and variable expression of stage-specific embryonic antigen, SSEA-4 [11].

Rapid tissue regeneration after tissue biopsy makes the gingiva an attractive tissue for cell isolation for therapeutic purposes, including the treatment of severe forms of aggressive CP [9]. However, the lack of deeper understanding of whether and how the inflammatory microenvironment affects the functional properties of GMSCs is reflected in recently published results in this field which are quite controversial. Some studies have demonstrated that GMSCs established from inflamed gingival tissues show similar phenotypes and minimal functional changes, compared to those obtained from healthy gingival tissues [12,13,14]. On the contrary, other studies have shown that such cells have altered phenotypic properties manifested by decreased FasL expression and have impaired immunomodulatory effects on T cells in vitro, compared to their healthy counterparts. In one study, inflamed GMSCs showed decreased colony-forming unit (CFU) efficiency and osteogenic differentiation along with increased adipogenic potential, compared to GMSCs from healthy gingiva [15]. There are also findings that GMSCs from inflamed gingiva proliferated faster than GMSCs from healthy gingiva, expressed a pro-fibrotic phenotype, but had reduced potential for osteogenic and adipogenic differentiation [16].

Regarding the possible influence of surrounding bacteria on GMSCs, Kang et al. 2019, studied the effect of *F. nucleatum* on GMSCs and found increased cell proliferation, migration and chemokine/cytokine production [17]. In contrast, LPS from *P. gingivalis* increased GMSCs proliferation but did not affect their CFU formation, osteogenic potential and inflammatory cytokine production [18]. Although such studies suggest that nearby bacteria and inflammatory microenvironment may exert different effects on the properties and function of GMSCs, available data is inconclusive and scarce. Therefore, we designed experiments to shed light on whether GMSCs isolated from inflamed human gingiva due to periodontitis (P-GMSCs) respond differently to *P. gingivalis* than their healthy counterparts (H-GMSCs).

## 2. Results

### 2.1. Establishment and Basic Characterisation of GMSC Lines

Ten H-GMSC lines were established from 10 donors with healthy gingiva as described in Materials and methods. Similarly, 12 P-GMSC lines were isolated from patients with CP. The initial assessment of gingival samples as healthy or inflamed was confirmed based on the absence or presence of abundant cellular infiltrate composed of polymorphonuclear and mononuclear cells (lymphocytes, plasma cells and macrophages) (data not shown). GMSC lines were propagated as bulky cultures and used in different experiments. Morphological characterisation showed that both types of GMSCs exhibited typical fibroblastoid morphology and adherence to plastic (Figure 1A). However, P-GMSCs formed larger CFU-fibroblasts (F) colonies (Figure 1B), they were larger in size (27.9 ± 8.5 µm) than H-GMSCs (16.8 ± 3.5 µm) (*p* < 0.001) as measured after their detachment (Figure 1C). They also proliferated faster up to the 5th passage, as estimated by the shorter population doubling time (PDT) (Figure 1D).

### 2.2. Differentiation Potential of GMSC Lines

H-GMSC and P-GMSC lines had the potential to differentiate into osteogenic, chondrogenic and adipogenic cells in vitro (Figure 2A). Although there was variability in the differentiation potential within both types of GMSCs lines, they both showed high osteogenic potential, moderate chondrogenic differentiation, whereas adipogenic differentiation was the lowest. Semiquantitative analysis showed that osteogenic differentiation of P-GMSCs was higher compared to H-GMSCs (*p* < 0.05) and that no differences between lines in terms of differentiation potential of the other two cell types were observed (Figure 2A). Higher osteogenic potential of P-GMSCs was further confirmed by higher gene expression of runt-related transcription factor 2 (RUNX2), bone morphogenetic protein-2 (BMP-2) and collagen type I alpha 1 chain (COL1A1) (Figure 2B), higher alkaline phosphatase (ALP) activity (Figure 2C) and higher expression of osteopontin (OPN) at the protein level (Figure 2D). No significant differences in osteocalcin (OCN) expression were observed at both mRNA and protein levels (data not shown).

### 2.3. Phenotypic Characteristics of GMSC Lines

H-GMSC and P-GMSC lines demonstrated high expression levels of MSC-associated cell markers such as CD90, CD73, CD44, CD73, CD105 and CD166. The expression of CD146 was moderate, whereas the expression of other markers (STRO-1 and SSEA4, including pericyte markers (PDGFR and NG2), was restricted to relatively small percentages of cells. There were no differences in any marker expression between H- and P-GMSCs. (Figure 3). Both types of these cell lines did not express hematopoietic cell markers: CD45, CD14 and CD34.

### 2.4. Modulation of Peripheral Blood Mononuclear Cell (PBMC) Proliferation by GMSC Lines

GMSCs have been known to suppress the proliferation of lymphocytes. Consequently, we co-cultured H- and P-GMSC lines with allogeneic human PBMCs stimulated with phytohemagglutinin (PHA) to evaluate if there was a difference in their immunosuppressive potential. Unstimulated PBMCs were used as controls. To arrest the proliferation of GMSCs, the lines were treated with mitomycin C. The results are presented in Figure 4A. H-GMSCs showed dose-dependent inhibition of PHA-induced proliferation of PBMCs (cell ratios 1:5 and 1:2 showed statistically significant differences; *p* < 0.05 and *p* < 0.001, respectively). The immunosuppressive effect of P-GMSCs was lower and was observed only in the 1:2 ratio (*p* < 0.001 compared to H-GMSCs). The *P. gingivalis* treatment of GMSCs for 24 h prior to co-cultures, further abrogated the inhibitory effect of H-GMSCs (*p* < 0.01), but not P-GMSCs.

### 2.5. Effect of P. gingivalis on the Production of IL-6 and IL-8 by GMSC Lines

GMSC lines were treated with *P. gingivalis* for 24 h. IL-6 and IL-8 levels were analysed in culture supernatants. At the basal level, P-GMSCs produced more IL-8 than H-GMSCs (*p* < 0.05). Both H- and P-GMSC lines produced significantly higher levels of IL-6 (*p* < 0.01) and IL-8 (*p* < 0.05) after *P. gingivalis* treatment (Figure 4B,C).

**Figure 3 ijms-23-03510-f003:**
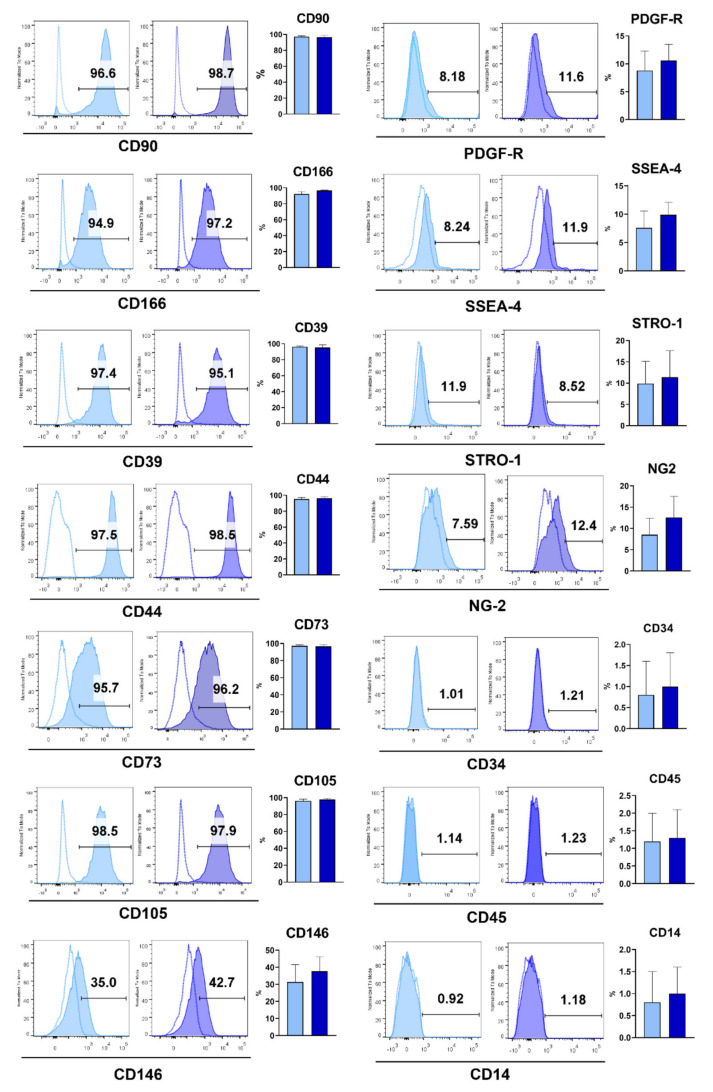
Expression of MSC markers on H- and P-GMSC lines. The methodology for cell staining and analysis is described in detail in Materials and methods. Histograms are representative from one of each line. The bars are placed according to the controls. Vertical columns represent the mean percentage of positive cells ± SD; *n* = 8 (H- or P-GMSC lines).

**Figure 4 ijms-23-03510-f004:**
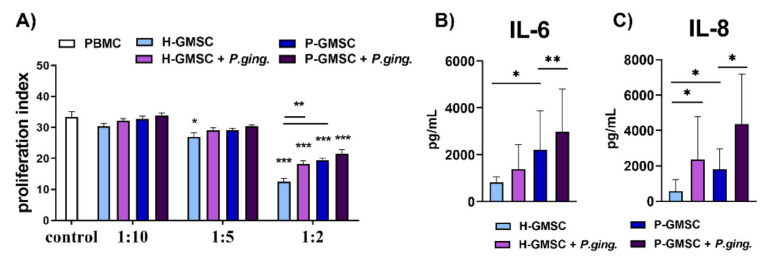
(**A**) Effect of untreated or *P. gingivalis* treated GMSC lines on PBMC proliferation. The co-culture experiments using different GMSC: PBMC ratios (1:10; 1:5; and 1:2) are described in detail in Materials and methods. Values are given as the mean proliferation index ± SD of four different experiments. * *p* < 0.05; *** *p* < 0.001 compared with control. ** *p* < 0.01 compared with corresponding cultures as indicated by bars. (**B**,**C**) Effect of *P. gingivalis* on the production of IL-6 and IL-8 by GMSC lines. The lines were stimulated with *P. gingivalis* for 24 h as described in Materials and methods. Untreated lines served as controls. Values are given as mean concentrations of cytokines in supernatants ± SD (*n* = 7–9). * *p* < 0.05; ** *p* < 0.01 compared with corresponding controls as indicated by bars.

### 2.6. Expression of Genes Associated with Inflammation in GMSCs Treated with P. gingivalis

H-GMSCs and P-GMSCs were incubated with *P. gingivalis* for 24 h, as described in Materials and Methods. The expression of several genes associated with inflammation was evaluated by qPCR (Figure 5). Although interindividual variability was seen, statistical analysis demonstrated that treatment with *P. gingivalis* led to up-regulation of Toll-Like Receptor (TLR)-2 in P-GMSCs (*p* < 0.05), whereas no significant differences in expression of TLR-3 and TLR-4 were seen. *P. gingivalis* up-regulated the expression of IL-6, Monocyte Chemoattractant Protein (MCP)-1 and Growth-Related Oncogene (GRO)-α in both H-GMSCs and P-GMSCs compared to untreated cells (*p* < 0.05). However, its effect on the expression of IL-6 and MCP-1 in P-GMSCs was stronger compared to H-GMSCs (*p* < 0.05). IL-8 and Regulated upon Activation Normal T Cell Expressed and Presumably Secreted (RANTES) were up-regulated only in P-GMSCs. Basal expression of IL-8 was also higher in P-GMSCs compared with H-GMSCs (data not shown).

### 2.7. Expression of Genes Associated with Immunomodulation in GMSCs Treated with P. gingivalis

Four genes associated with immunomodulation have been studied. After treatment of lines with *P. gingivalis*, the expression of Interferon (IFN)-γ-Induced Protein 10 kDa (IP-10) was significantly higher in P-GMSCs, compared with H-GMSCs, in contrast to the expression of Indoleamine 2, 3-Dioxygenase 1 (IDO1) which was down-regulated. Treatment with *P. gingivalis* did not change the expression of Transforming Growth Factor (TGF)-β in any of the cell line types. However, the expression of Latency-Associated Peptide (LAP) tended to be higher in some P-GMSC lines after *P. gingivalis* treatment, resulting in a decrease in the TGF-β/LAP ratio (Figure 6).

### 2.8. Expression of Genes Associated wit Tissue Regeneration/Repair in GMSCs Treated wit P. gingivalis

Seven genes associated with tissue regeneration/repair in GMSCs after treatment with *P. gingivalis* were analysed. Of these, only Hepatocyte Growth Factor (HGF) was down-regulated in H-GMSCs both at the basal level (data not shown) and after *P. gingivalis* treatment. In contrast, *P. gingivalis* up-regulated HGF expression in P-GMSCs and the difference was statistically significant (*p* < 0.05) compared with H-GMSCs. Four genes, Hypoxia-Inducible Factor (HIF)-1α, Stromal cell–Derived Factor (SDF)-1, Osteoprotegerin (OPG) and Matrix Metalloproteinase (MMP)-3, were up-regulated only in P-GMSCs after *P. gingivalis* treatment (*p* < 0.05, compared with untreated P-GMSCs). In contrast, Insulin-like Growth Factor Binding Protein 4 (IGFBP4) and Tissue Inhibitor of Metalloproteinase (TIMP)-1 were non significantly changed in both H- and P-GMSC lines (Figure 7).

## 3. Discussion

The main goal of our study was to test the hypothesis that GMSCs originating from healthy and inflamed (caused by periodontitis) gingiva respond differently to *P. gingivalis*. To prove or reject this hypothesis, it was necessary to establish GMSC lines representative for research. Therefore, we established H-GMSC lines (*n* = 10) from donors with healthy gingiva and P-GMSC lines (*n* = 12) from patients with CP. We showed that H- and P-GMSC lines shared similarities in adherence to plastic, fibroblastoid morphology and phenotype. These characteristics were in accordance with the criteria established by the International Society of Cellular Therapy (ISCT) for MSCs, including high expression of CD105, CD73 and CD90, and the absence of hematopoietic markers [19]. Additionally, MSCs express high or variable levels of other markers including CD29, CD44, CD56, CD166, CD146, CD271, STRO-1 and SSA4 [20], which was also confirmed in our study for both types of GMSCs. The source of GMSCs is lamina propria where a network of small blood vessels is present [21] which is in agreement with our previous findings that GMSCs also express characteristic markers of pericytes, such as NG2 and PDGFR [22]. We have demonstrated that the proliferation rate of GMSCs is high, which is consistent with other publications [23,24]. P-GMSCs had a much higher proliferation up to the 5th passage, as judged by decreased doubling time and formation of larger CFU-F, compared with H-GMSCs.

MSCs possess trilineage differentiation potential into osteoblasts, adipocytes and chondroblasts [19]. Most P-GMSC lines manifested higher osteogenic potential compared to H-GMSCs, in contrast to moderate chondrogenic and low adipogenic potentials exhibited by both lines. Reports of GMSCs’ response to inflammation are controversial. Our results are comparable to those published by Tomasello et al. 2017 [14], but they differ from observations made by Ge et al. 2012, who demonstrated similar CFU-F capacity and a slight reduction in the population doubling time of P-GMSCs compared with H-GMSCs [25]. One study showed that GMSCs from the inflammatory microenvironment had lower CFU efficacy, increased adipogenic ability, and decreased osteogenic potential compared with GMSCs established from healthy gingiva [15]. In another study, GMSCs from inflamed gingiva had increased proliferative activity and pro-fibrotic properties but showed reduced osteogenic and adipogenic potentials [16]. We have demonstrated that the adipogenic potential of both H- and P-GMSCs is relatively low, as shown by other authors [25]. The reason for these variabilities is not clear. These discrepancies may be due to differences in tissue donors, procedures for cell isolation or general culture conditions.

Constant exposure of GMSCs to microbial flora during periodontitis could significantly modify their regenerative and tissue homeostatic potentials. Under inflammatory conditions, GMSCs are mobilised towards the site of infection or damage and come in close contact with bacteria and their products [26]. *P. gingivalis* comprise 85.75% of subgingival plaque samples from patients with CP, which makes it the most abundant amongst all periodontopathic bacteria [6]. That was the reason why we chose to study the interaction between *P. gingivalis* and GMSCs, assuming that the response of H-GMSCs and P-GMSCs, in terms of activation of 19 genes involved in inflammation, cell proliferation, tissue regeneration and tissue remodelling, is different. The treatment of H-GMSCs with *P. gingivalis* mimics initial contact with this species of bacteria, whereas the use of P-GMSCs is relevant for CP. *P. gingivalis* can directly destroy periodontal tissues by secreting toxic and virulent factors. Virulence factors present on fimbria and capsule that play the most important role in the colonisation of the host tissues by *P. gingivalis* are: lipopolysaccharide (LPS), lipoproteins, lipoteichoic acids, haemagglutinins, gingipains, outer membrane proteins and outer membrane vesicles. This species of Gram-negative bacteria also triggers the local immune response by binding their pathogen-associated molecular pattern (PAMPs) to corresponding pattern recognition receptors (PRRs) expressed on host cells. These receptors have a role in promoting the innate immune response, which is mainly described as pro-inflammatory [27]. The most important amongst them are TLR 4, an LPS receptor, and TLR-2, a receptor for lipoproteins of *P. gingivalis* [28,29]. MSCs also express TLR-3, which we also documented in this study. However, it is not known how signalling through different TLRs modulates the course of periodontitis [30].

We found that P. gingivalis predominantly stimulated TLR-2 expression over TLR 4 expression in most H-GMSC lines. However, statistical analysis showed that only TLR-2 expression was up-regulated in P-GMSC. In contrast, the stimulatory effect on TLR-4 and TLR-3 expression was observed only in some P-GMSC lines, but the differences in the group as a whole were not statistically significant. Simultaneous activation of TLR-2 and TLR-3 has been shown to synergistically trigger the production of IL-6, IL-8 and MCP-1 in human periodontal ligament stem cells (PDLSC) [31]. In addition, P. gingivalis has been shown to stimulate the secretion of proinflammatory cytokines/chemokines (IL-1β, TNF-α, IL-6, IFN-γ, IL-8, RANTES and MCP-1), in human peripheral blood ex vivo. Cytokine levels were dependent on strain and dose of *P. gingivalis*. Although cytokine profiles amongst donors were comparable, great interindividual variability was found [32].

The results of our study revealed that *P. gingivalis* up-regulation of IL-6, IL-8, MCP-1, GRO-α, and RANTES was more potent in P-GMSCs compared with their healthy counterparts. IL-6, MCP-1, and GRO-α were up-regulated in H-GMSCs. GRO-α and IL-8 are potent chemoattractants for neutrophils that form the first line of defence in the gingival sulcus. On the other hand, MCP-1 recruits monocytes/ macrophages, while RANTES is a chemokine for T cells and both chemokines form the second line of defence [33]. It is currently unclear whether the stronger pro-inflammatory response of P-GMSCs to *P. gingivalis* is due to synergistic signalling through multiple TLRs or whether other PAMPs are also involved.

Our results suggest that GMSCs isolated from periodontitis tissue have already been primed to respond better to *P. gingivalis*, as assessed by up-regulation of proinflammatory genes involved in CP progression. The effect of *P. gingivalis* depends on the type of MSCs and the receptors involved. For example, dental follicle progenitor cells (DFPCs) expressed TLR2 and TLR4 at both mRNA and protein levels, but their treatment with *P. gingivalis* LPS had no effect on proinflammatory cytokines expression [34,35]. Contrary to the results of the current study, we showed in our previous work that treatment of DFPCs with a TLR4 agonist increased their suppressive potential [36]. TLR4 stimulation has been shown to polarise MSCs into the pro-inflammatory MSC1 phenotype, while signalling via TLR3 transforms MSCs into the immunosuppressive MSC2 phenotype [7,37]. Based on the expression of several genes involved in inflammation, it can be assumed that *P. gingivalis* favours the development of the GMSC1 type, and this process is more prominent in P-GMSCs.

GMSCs have been shown to exhibit immunosuppressive capability in a number of experimental models both in vitro and in vivo [9,38,39]. This phenomenon was confirmed in our study in a model of PHA-induced PBMC proliferation. The inhibitory effect of P-GMSCs was lower than H-GMSCs, suggesting again that less immunosuppressive P-GMSCs resembling MSC1 type, retain their, pro-inflammatory memory” in culture. This finding is consistent with higher basal production of pro-inflammatory mediators (IL-6 and IL-8) in culture supernatants of P-GMSCs and higher basal expression of IL-8 mRNA compared to H-GMSCs. Suppression of PBMC proliferation induced by GMSCs is mediated by many immunosuppressive factors such as IL-10, IDO or inducible nitric oxide synthase (INOS), whose production has been triggered by IFN-γ in co-culture [10]. IDO is an intracellular monomeric, hem-containing enzyme that controls the breakdown of tryptophan, one of the important essential amino acids, to kynurenine. Its effect is predominantly immunosuppressive and pro-tolerogenic due to tryptophan depletion [40]. IDO is not constitutively expressed by MSCs, but may be significantly induced by inflammatory mediators, such as IFN-γ [10,41,42]. Activation of the IFN-γ-IDO pathway in MSCs is mediated through up-regulation of B-H1 [43] or via the JAK/STAT1 signalling pathway [44]. The finding in our study that *P. gingivalis*-treated GMSCs (especially P-GMSCs) lost significantly the potential to inhibit PHA-induced proliferation of PBMCs is consistent with the findings that IDO expression was down-regulated in P-GMSC lines compared with *P. gingivalis*-treated H-GMSCs. In addition, P-GMSCs may have reduced immunosuppressive potential in PBMC cultures, which is further potentiated by *P. gingivalis*, due to their lower ability to produce IL-10 and induce differentiation of T regulatory cells (Tregs). Such a result was published for periodontal ligament derived MSCs isolated from periodontitis using the same co-culture model, suggesting that a reduction in immunosuppressive mechanisms is needed to promote acute inflammation [45].

One of the genes associated with IFN-γ is IP-10. IP-10, a C–X–C motif chemokine 10 (CXCL10) produced in response to IFN-γ, binds to the CXCR3 receptor. IP-10 induces chemotaxis, apoptosis, cell growth (stimulation or inhibition depending on whether it binds to CXCR3-A or CXCR3-B, respectively) and angiostasis (inhibition of angiogenesis) [46]. IP-10 recruits inflammatory T helper 1 (Th1) cells into inflamed gingival tissue [47], but its exact role (pro- or anti-inflammatory) has not yet been elucidated. Gingipain-R from *P. gingivalis* has been shown to stimulate IL-8 production, but decrease IP-10 in human gingival fibroblasts upon contact with T cells [48]. In another study, *P. gingivalis* did not induce IP-10 expression from neutrophils, PBMCs, or gingival epithelial cells. Moreover, *P. gingivalis* inhibited the release of IP-10 stimulated by IFN-γ [49]. This finding has been confirmed in most of our H-GMSC lines. However, *P. gingivalis* significantly up-regulated the expression of IP-10 in P-GMSCs, suggesting that its role in initiating gingivitis and maintaining CP may be different. Our findings in P-GMSCs support the view that IP-10 is rather a pro-inflammatory and angiostatic chemokine in CP, but this hypothesis deserves to be tested in the next experiments.

One of the main immunosuppressive cytokines is TGF-β. However, it is a multifunctional cytokine that controls proliferation, cell differentiation, apoptosis, angiogenesis, and immune responses. TGF-β is secreted by inflammatory cells and Tregs during inflammation and tissue injury, and regulates tissue remodelling by stimulating the production of connective tissue components and matrix proteins by fibroblasts and other cells, including MSCs [50]. The secreted TGF-β is tightly bound to a latent complex consisting of latency-associated peptide (LAP), and latent TGF-β-binding protein (LTBP). In this TGF-β/LAP/LTBP complex, LAP confers latency of TGF-β, while its sequestration into the extracellular matrix (ECM) and conversion from the latent form to the active form is mediated by LTBP [51]. In our GMSCs cultures, TGF-β was not significantly changed after treatment with *P. gingivalis*. However, up-regulation of LAP in most P-GMSCs cultures suggests that TGF-β is sequestered in ECM during the chronic phase of periodontitis. Nevertheless, this hypothesis should be viewed in the context of other mechanisms where inflammatory, anti-inflammatory, destructive and reparative processes are strictly controlled.

Apart from their role in host defence and inflammation, MSCs are widely recognised as key players in tissue regeneration [26]. In this context, secreted products or extracellular vesicles (EV) of GMSCs have been shown to enhance the regeneration of inflamed periodontal tissue in a number of experimental models in mice [9]. For this reason, we analysed the influence of *P. gingivalis* on the expression of various biomolecules in GMSCs involved in cell proliferation, tissue regeneration or tissue remodelling.

HIF-1α induces the transcription of more than 60 genes, including vascular endothelial growth factor (VEGF) which is involved in angiogenesis and helps to increase oxygen delivery to hypoxic regions. It has been shown that human PDLSCs infected with *P. gingivalis* up-regulated HIF-1α, which is involved in the accumulation of reactive oxygen species (ROS), and the production of pro-inflammatory cytokines [30]. In addition, HIF-1α promotes MSC migration to ischemic and hypoxic sites by up-regulating the expression of different biomolecules, such as SDF 1, also known as CXCL12 chemokine. MSCs, including human PDLSCs and gingival fibroblasts, constitutively express the SDF 1 receptor (CXCR4) which engagement upon CXCL12 ligation plays a key role in tissue repair, bone remodelling, MSC proliferation, as well as in a number of physiological and pathological processes, including modulation of inflammation [52].

SDF 1 promotes the recruitment and proliferation of MSCs, increases their osteogenic differentiation [53], promote angiogenesis [54] and reduces inflammation [55]. We obtained interesting results showing that *P. gingivalis* did not induce the expression of both HIF-1α and SDF-1 in H-GMSCs, but significantly increased their expression in P-GMSCs. These results indicate that the HIF-1α—SDF-1– CXCR4 axis is not involved in the initiation of gingivitis. However, this signalling pathway may be relevant for the late stage of periodontitis when the need for tissue regeneration/repair is much more necessary.

HGF is a pleiotropic, paracrine factor involved in tissue regeneration [56], proliferation and migration of epithelial and endothelial cells [57]. MSCs secrete HGF under inflammatory stimuli [58]. For the first time, we showed that HGF is the only one of the investigating genes that was silenced at the basal level in GMSCs and did not respond to *P. gingivalis* in H-GMSCs. Activation of the HGF gene was observed in P-GMSC lines, suggesting again that P-GMSCs in the inflammatory microenvironment have already been primed to activate the HGF gene in response to *P. gingivalis*. The exact role of HGF in the course of periodontitis is not yet clear. In addition to its possible role in tissue regeneration, HGF could have other functions in combination with other factors. For example, HGF together with SDF-1, MCP-1, IL-6 and VEGF promote angiogenesis and survival of endothelial cells, which is important during the regeneration of damaged periodontal tissue [59]. HGF is involved in immunosuppression by promoting the development of Tregs [56]. Therefore, downregulation of HGF in H-GMSCs may be important for the promotion of inflammation in the early phase of periodontitis. In contrast, its activation in P-GMSCs may be more relevant for regenerative processes, but its action should be viewed in the context of activation of other genes involved in the pathophysiology of periodontitis [59].

IGFBP4 plays an important role in bone metabolism and participates in senescent-associated processes in MSCs [60,61]. We showed that IGFBP4 expression in both H- and P-GMSCs was not modified by *P. gingivalis*. A possible explanation for this observation could be that cell lines in our study were in early passages when the senescence processes in them are silenced. Unlike IGFBP4, OPG is up-regulated in the presence of *P. gingivalis* in P-GMSCs, suggesting that GMSCs chronically stimulated with *P. gingivalis* during periodontitis may be a target for promoting blastogenesis and bone repair. OPG is known to function as a soluble decoy receptor for the receptor activator of nuclear factor-kappa B ligand (RANKL). By blocking RANKL, osteoclastogenesis triggered by RANKL-RANK interaction is suppressed and osteoblastogenesis is induced [62]. In these processes, locally produced OPG is more important for osteogenic regeneration than its circulating form [63] and this phenomenon could be relevant for our study. The role of GMSCs in osteoblastogenesis independent of the RANKL/OPG system has been confirmed recently. CD39 produced from GMSCs has been shown to stimulate their osteogenic capacity through the Wnt/β-catenin pathway [64]. However, some other contradictory results have been published. Reddy et al. [65] have shown that LPS and gingipains from *P. gingivalis* up-regulate RANKL and down-regulate OPG mRNA expression and protein production through prostaglandin E2 (PGE2) secretion. LPS from *P. gingivalis* has also been shown to inhibit osteoblastic differentiation and mineralisation in PDLSCs involved in periodontal tissue regeneration [66]. In this context, it is obvious that further experiments are needed to examine how *P. gingivalis* treatment of GMSCs, established from healthy versus inflamed tissue, modifies the expression of RANKL and the RANKL/OPG ratio, and to clarify whether osteogenic differentiation of these cells is activated, suppressed or not significantly changed.

MMP-3 is a member of the MMP family, which consists of 23 endopeptidases involved in the degradation of extracellular matrix (ECM) components. Tissue inhibitor of metalloproteinase (TIMP) is a natural MMP inhibitor. The balance between MMPs and TIMPs is essential for the integrity of the ECM, while the course of many diseases can be determined by proteolytic changes in which these biomolecules participate [67]. In addition to CP, MMP-3 is involved in several physiological and pathological conditions, such as accumulation of inflammatory cells, promotion of vascular invasion and osteoclast differentiation, degradation of cartilage matrix and inhibition of MSC differentiation. On the other hand, TIMP-1 has an antagonistic function against MMP-3. Under the physiological condition, there is a balance between MMPs and TIMP-1, and any increase in TIMP-1 with or without a decrease in MMP can lead to an increase in ECM [68]. Following the *P. gingivalis* treatment in our study, neither MMP-3 nor TIMP-1 expression in H-GMSCs was modified, whereas MMP-3 and MMP-3/TIMP-1 ratio was significantly increased in most P-GMSC lines, all of which suggests that pathological degradation of ECM within periodontal tissues is more pronounced in the advanced stage of periodontitis. Therefore, based on the analysis of this set of genes in P-GMSCS, it could be assumed that *P. gingivalis* dysregulates regenerative and reparative processes during CP rather than stimulating them.

In conclusion, our results showed that GMSCs established from CP were already primed to respond stronger to *P. gingivalis* in vitro than their healthy counterparts. In addition to their faster proliferation, better osteogenic potential and lower immunosuppressive capability under basal conditions, *P. gingivalis* treatment affected numerous genes of pro-inflammatory cytokines/chemokines and genes involved in tissue destruction/regeneration differently in P-GMSCs compared with H-GMSCs. Cumulatively, all presented results suggest that P-GMSCs retain their MSC1-like properties ex vivo, which are further significantly enhanced by *P. gingivalis* treatment. Further studies are needed to address whether and how *P. gingivalis* changes the functional properties of GMSCs in terms of their proinflammatory versus immunosuppressive qualities.

## 4. Materials and Methods

### 4.1. Tissue Donors and General Study Design

This was a collaborative study conducted at the Medical Faculty Foča (MMF), University of East Sarajevo, Bosnia and Herzegovina (Clinical and laboratory part of the research), Clinic for Dentistry (Department for Oral Surgery), Military Medical Academy (MMA), Belgrade, Serbia (Clinical part of the research) and the Institute for the Application of Nuclear Energy Research (INEP), University of Belgrade, Serbia (Laboratory part of the study). The study was conducted in accordance with the Declaration of Helsinki, and the protocol was approved by the Ethics Committee of MMA (permission number: 02-07/01/2019) and the Ethics Committee of Medical Faculty Foča (permission number: 01-2-4/06.10.2020.). Gingival tissue samples were collected after the written informed consent was obtained from the donors. Clinically healthy gingival samples were collected from subjects who had no history of periodontal disease. Samples of healthy gingival tissue were taken from 10 donors (6 males and 4 females; age range 36–52 years). Three of the donors were from the Department of Dentistry, MFF, while six were from the Clinic for Dentistry, MMA.

The periodontitis group consisted of 12 donors (four patients were from the Department of Dentistry, MFF and 8 patients were from the Clinic for Dentistry, MMA. Periodontitis was diagnosed according to the American Academy of Periodontology (AAP) Classification of Disease. Staging and gradation of the disease were performed according to the Consensus report of workgroup 2 of the 2017 World Workshop on the Classification of Periodontal and Peri-Implant Diseases and Conditions [69]. All patients were classified as stage II based on the stage of CP. They had an interdental clinical attachment loss (CAL) of 3–4 mm at the site of greatest loss; had no tooth loss and the maximum probing depth was <5 mm. All patients were classified as periodontitis grade A (slow rate of progression). There were 6 males and 6 females (ages 42–55 years.). Apart from matching by age and sex, subjects with healthy gingiva and patients with periodontitis had no diabetes, systemic autoimmune diseases, or malignancies and were not smokers. In addition, they did not receive antibiotics or immunosuppressive drugs for one month before tissue sampling. Other data from the medical history of study participants were not recorded. Periodontitis-inflamed tissue was collected during the flap debridement procedure, whereas healthy gingival specimens were taken during tooth extraction for orthodontic purposes or during the removal of hard dental plaque.

The initial healthy and inflamed gingival tissue was subjected to classical pathohistological analysis. Tissue sections were stained with hematoxylin and eosin (H&E) (Sigma-Aldrich, Darmstadt, Germany) and analysed under a light microscope (Olympus, Hamburg, Germany). Evaluation of gingival samples as healthy or inflamed was confirmed based on the absence or presence of abundant cellular infiltrate composed of polymorphonuclear and mononuclear cells (lymphocytes, plasma cells and macrophages).

### 4.2. Establishment of GMSC Lines

The method for isolating human gingival cells and establishing H- and P-GMSC lines was a modified procedure that has already been published [25,70]. Gingival samples were washed in phosphate-buffered saline (PBS), de-epithelialised with a scalpel to remove most of the epithelial cells, minced and enzymatically digested with collagenase type II (5 µg/mL) and DNAse (40 IU/mL) in serum-free α-MEM (all from Sigma-Aldrich, Darmstadt, Germany), for 2 h at 37 °C and 5% CO_2_ in a humidified atmosphere. After the incubation period, the tissue was gently pressed through a 30 µm nylon mesh using a plunger from a sterile syringe, then it was rinsed with α-MEM medium and centrifuged at 1800 rpm for 10 min. The obtained primary gingival cells were placed in 6-well cell culture dishes (Sarstedt, Numbrecht, Germany) at a density of 10,000 per cm^2^ and cultured in a complete MSC medium consisting of α-MEM with 10% FCS and 100 IU/mL of penicillin, 50 μg/mL streptomycin, 2.5 μg/mL amphotericin B (all from Thermo Fisher Scientific, Dreieich, Germany), 1% sodium pyruvate and 100 µM L-ascorbate-2-phosphate (both from Sigma-Aldrich, Darmstadt, Germany). Non-adherent cells were removed by changing the complete culture medium twice a week. After reaching 70% confluence, GMSCs were passaged by incubating cells with 0.02% trypsin/0.02% Na EDTA (Sigma-Aldrich, Darmstadt, Germany) in PBS. The cells were then harvested and washed in a culture medium and plated in cell culture dishes at a density of 5000 per 1 cm^2^. To establish bulk-cultured GMSCs lines, cell suspensions were seeded into 25-cm^2^ tissue culture flasks in a complete α-MEM medium without amphotericin B and used for further experiments. The size of GMSCs from the 4th passage was measured after their detachment. The suspension of rounded cells was placed on microscopic slides, observed under an inverted microscope (Olympus, Hamburg, Germany), and photographed using an attached camera. Images were analysed with ImageJ software. The diameter of individual cells was measured manually, and the values were presented in µm according to the scale.

### 4.3. Colony Formation Assay

CFU fibroblasts (CFU-F) assay was performed after the third passage of GMCS lines. GMSCs (*n* = 3) of each H- and P-GMSCs were seeded at 400 cells per well in a six-well plate and cultured in a complete α-MEM medium in a cell incubator (37 °C, 5% CO_2_) as described above. The culture medium was changed twice a week. After two weeks, GMSCs cultures were washed with PBS, fixed with 4% paraformaldehyde (Sigma-Aldrich, Darmstadt, Germany) and stained with H&E. Colonies (cell aggregates of ≥50 cells), recognised as CFU-F, were analysed and calculated under an inverted microscope (Olympus, Hamburg, Germany). The characteristic images were taken with a camera attached to the microscope.

### 4.4. Population Doubling Time Assay

Population doubling times (PDT) for GMSCs cultures were determined as described [69,71] with a slight modification. Third, fifth and eighth generations of H-GMSCs and P-GMSCs were trypsinised and cultured in 24-well plates (1.5 × 10^4^ cells/mL) at 37 °C in a cell incubator containing 5% CO_2_ as described for general GMSCs cultures. Wells (*n* = 3) were randomly selected, trypsinised and the cell number was calculated manually using a Neubauer chamber (each well was calculated twice). The number of cells was determined daily for up to 7 days. PDT was calculated using the following formula: PDT = t × lg2/(lgNt-lgNo), where t indicates incubation time; No represents initial cell number and Nt refers to the number of cells in culture of t hours. PDT was calculated for four H-GMSC and four P-GMSC lines and mean values ± SD were determined.

### 4.5. In Vitro Differentiation Capacity

In order to determine osteogenic differentiation (OD) we plated H- and P-GMSCs at a density of 5 × 10^4^ cells in six-well plates with inserted plastic coverslips until the monolayer reached confluence. Cells were subsequently cultured for 21 days in an osteogenic inductive medium consisting of the complete α-MEM medium supplemented with 10 nM dexamethasone (Galenika, Belgrade, Serbia), 10 mM glycerophosphate and 0.05 mM ascorbic acid (both from Sigma-Aldrich, Darmstadt, Germany). The medium was changed twice a week and coverslips were washed with PBS at the end of the cultivation period. Thereafter we fixed cells with 4% paraformaldehyde for 60 min at room temperature, washed them twice with distilled water, stained them with 2% Alizarin Red (Sigma-Aldrich, Darmstadt, Germany) for 45 min in the dark, and finally washed them again with distilled water and PBS. Coverslips were mounted on microscopic slides. The expression of the RUNX2 gene was determined after 7 days, whereas the expression of other osteogenic genes (BMP-2, COL1A1, OPN and OCN), OPN and OCN proteins (flow cytometric analysis) and ALP activity was performed after 14 days of osteogenic differentiation.

To evaluate adipogenic differentiation (AD), confluent cell monolayers were cultured for 21 days with the complete α-MEM medium supplemented with 0.5 µM dexamethasone (Galenika, Belgrade, Serbia), 0.5 µM isobutyl-methylxanthine (IBMX), (Sigma-Aldrich, Darmstadt, Germany) and with 50 µM indomethacin (R&D Systems, Minneapolis, MN, USA). The procedure of cultivation, washing and fixation was the same as described for OD. Coverslips were washed with 60% isopropanol for 5 min and stained with 0.3% Oil Red O (Sigma Aldrich). Finally, the coverslips were washed with tap water and stained with hematoxylin for 1 min, washed again with tap water and mounted on microscopic slides.

For chondrogenic differentiation (CD) assessment, 5 × 10^5^ cells were pelleted in Eppendorf tubes by centrifugation at 1,800 rpm for 10 min. The pellet was cultivated in the complete α-MEM medium with the addition of the chondrogenic supplement containing TGF-β3 (10 ng/mL) (R&D Systems, Minneapolis, MN, USA), dexamethasone (100 nM) (Galenika, Belgrade, Serbia),) and 2- 16hosphor-L-ascorbic acid (50 ng/mL) (Sigma-Aldrich, Darmstadt, Germany). The pellets were cultured for 21 days. At the end of the cultivation period, the pellets were cryopreserved in an embedding medium (Bio-Optica, Milan, Italy) and stored at −80 °C for further cryostat sectioning (Leica Biosystems, Barcelona, Spain). Cryostat sections were air-dried and fixed with 4% paraformaldehyde, rinsed with distilled water and stained with Alcian blue solution (Sigma-Aldrich, Darmstadt, Germany) for the next 45 min at room temperature. After staining, sections were washed twice with distilled water and counterstained with 0.1% Nuclear Fast Red solution (Sigma-Aldrich, Darmstadt, Germany). Negative controls in all differentiation protocols were GMSCs cultured in the complete basal α-MEM medium.

The osteogenic, adipogenic and chondrogenic cultures were observed under a light optical microscope (Olympus, Hamburg, Germany). All images were analysed offline in ImageJ software (National Institutes of Health, Bethesda, MD, USA). Semiquantitative analysis for all differentiated cells was as follows: Index 0: no visible positivity; Index 1: Mild positivity of individual cells and the presence of 1 to 2 smaller mineralised islets (OD) or 1–2 positive cells (AD and CD) in at least one of the 10 analysed microscopic fields; Index 2: Mild positivity of individual cells and the presence of up to 5 small and medium-sized mineralised islets (OD) or up to 5 positive cells (AD and CD) in at least two of the 10 analysed microscopic fields; Index 3: Presence of up to 10 mineralised islets of different sizes (OD) or up to 10 positive cells (AD and CD) in at least 5 of the 10 analysed microscopic fields; Index 4: Presence of mineralised nuclei of all sizes, some merged (OD) or most positive cells (AD and CD) on all 10 analysed microscopic fields.

### 4.6. Flow Cytometry Analysis

The flow cytometry analysis of H-GMSCs and P-GMSCs was performed on a BD LSR II flow cytometer (BD Biosciences, Franklin Lakes, NJ, USA) by staining the cells with following monoclonal antibodies at dilutions recommended by the manufacturer: IgG1 isotype control-FITC (MOPC-21), IgG1 negative control-PE (MOPC-21), IgG1 negative control-APC (MOPC-21), IgG1 negative control-APCCy7 (MOPC-21), anti-CD14-FITC (63D3), anti-STRO-1-FITC (STRO-1), anti-CD45-APC (HI30), anti-CD90-PE (5E10), anti-CD73-biotin (AD2), anti-SSEA-4-biotin (MC-813-70), anti-CD166-PE (3A6), purified anti-NG2 (MEL62), anti-CD105-APC (43A3), streptavidin-APC, streptavidin APCCy7 (all from BioLegend, Basel, Switzerland); anti-PDGFR-Alexa Fluor 546 (D-6), anti-CD146-Alexa Fluor 488 (P1H12) (both from Santa Cruz Biotechnology, Dallas, Texas, USA); anti-CD39-FITC (eBioA1) (eBioscience, San Diego, CA, USA); anti-CD34-FITC (581) (Elabscience, Wuhan, China); anti-CD44-APCCy7 (IM7); anti-mouse IgG (whole molecule)-FITC (IgG fraction of antiserum) (Sigma-Aldrich, Darmstadt, Germany); rabbit anti-goat polyclonal antibody-Alexa Fluor 488 (Abcam, Cambridge, UK); anti-OPN (polyclonal goat antibody) and anti-OCN (190125) were purchased from R&D Systems (Minneapolis, MN, USA). The staining procedure was conducted in 2% FCS/0.01% NaN3 in PBS with 30 min incubation period at 4 °C. For each sample, doublets were excluded according to forward scatter (FSC)-A/FSC-H, with more than 5000 gated cells according to their specific FSC-A/side-scatter (SSC)-A properties. Signal overlaps between the channels were compensated before each experiment using single labelled cells, and non-specific fluorescence was determined by using the appropriate isotype control antibodies. The acquired data were analysed offline in FlowJoVX (BD Biosciences, Franklin Lakes, NJ, USA).

### 4.7. ALP Activity

ALP activity was measured using ScienCell assay (Sciencell Research Laboratories, Carlsbad, CA, USA), according to enclosed instructions. In brief, after 14 days of cultivation in a control or osteogenic differentiation medium, H- and P-GMSC were washed twice with PBS and harvested with lysis buffer and sonicated. After centrifugation at 14,000× *g* for 5 min, debris-free supernatants were used to determine levels of total protein content and ALP. Total protein content was measured using a Pierce™ BCA protein assay Kit (Thermo Fisher Scientific, Dreieich, Germany). BSA standard or samples were transferred to a 96-well plate to which working reagents was added (working reagent 50:1 ratio of assay reagents A and B). The plate was incubated for 30 min at 37 °C, before reading the absorbance at 570nm at the microplate reader (ELx800, BioTek, Winooski, VT, USA). To adjust the same protein content between samples, samples were diluted with the appropriate amount of assay buffer. Further, 50 µL of the sample was incubated with a 50 µL working concentration of substrate (5:1 ratio of assay buffer and substrate stock solution), for the next 60 min at 37 °C, in dark. The reaction was terminated with a stop solution before reading the absorbance at 405nm. The level of ALP activity in H- and P-GMSC cultivated in the osteogenic medium was normalized to control, non-treated cells (=1).

### 4.8. Cultivation of P. gingivalis

*P. gingivalis*, strain 33277, was purchased from the American Type Culture Collection (ATCC, Manassas, VA, USA). Cultivation of *P. gingivalis* was performed under strict anaerobic conditions (85% N_2_, 5% H_2_ and 10% CO_2_) at 37 °C in the Brain-Heart-Infusion (BHI) broth supplemented with menadione (1 μg/mL) and hemin (5 µg/mL), both purchased from Sigma–Aldrich. After harvest, *P. gingivalis* colonies were rinsed twice with PBS and resuspended in 1mL of PBS. The concentration of *P. gingivalis* was determined with a spectrophotometer at 600 nm by McFarland turbidity standard No. 0.5, where 0.1 OD refers to 1 × 10^8^ colony-forming units per mL (CFU/mL). To ensure the loss of viability of *P. gingivalis* prior to GMSCs treatment, *P. gingivalis* was being heat-inactivated by incubation at 70 °C for 30 min in addition to atmospheric oxygen exposure.

### 4.9. Treatment of GMSC Lines with P. gingivalis

H-GMSCs and P-GMSCs from the 5th passage were seeded in 6-well plates (Thermo Fisher Scientific, Dreieich, Germany) at a density of 5000 per cm^2^. After reaching confluence, GMSCs were incubated with *P. gingivalis* at a multiplicity of infection (MOI) of 100 bacteria per one GMSCs. Cultivation was continued for the next 24 h in a complete culture medium at 37 °C and 5% CO_2_ humidified atmosphere, as already described. Untreated GMSCs served as a control. At the end of incubation, cell-free culture supernatants were collected. GMSCs were washed twice with sterile PBS to remove bacteria and harvested by trypsinisation. Cell pellets were stored in Trizol reagent (Thermo Fisher Scientific, Dreieich, Germany) at −80 °C for the subsequent RNA extraction.

### 4.10. Real-Time Quantitative PCR

Total RNA was extracted from cultured cells using the Total RNA Purification Mini Spin Kit (Genaxxon Bioscience GmbH, Ulm, Germany) according to the manufacturer′s protocol. A High-Capacity cDNA Reverse Transcription Kit (Thermo Fisher Scientific, Dreieich, Germany) was used to transcribe 0.1 µg of isolated RNA as a template. The synthesised cDNA was then subjected to Real-Time Quantitative PCR (qPCR) analysis using a SYBR Green PCR Master Mix (Thermo Fisher Scientific, Dreieich, Germany) in a 7500 real-time PCR machine (Applied Biosystems, Waltham, MA, USA) under the following conditions: 10 min at 95 °C activation, 40 cycles of 15 s at 95 °C and 60 s at 60 °C. The results were normalised against β-actin for each sample and expressed as a relative target abundance (versus the non-treated sample of each donor) using the 2^−ΔΔCt^ method [72]. To compare differences in expression of each marker between H-GMSCs and P-GMSCs upon *P. gingivalis* treatment, the expression of each marker after *P. gingivalis* treatment was calculated for each donor as a fold change of basic level expression used as 1. To compare variances in basic expression levels of analysed markers between non-treated H-GMSCs and P-GMSCs, the expression of each marker on non-treated cells was calculated as fold change of marker expression in H-GMSCs of one donor used as 1. Primers used in the study are listed in Table 1. All primers were purchased from Thermo Fisher Scientific.

### 4.11. Co-Culture Proliferation Assay

The capacity of H- and P-GMSCs to stimulate human PBMC proliferation was evaluated in co-culture, as we previously described [36]. GMSCs (1 × 10^4^/well) were seeded in flat-bottom 96-well plates in complete RPMI-1640 medium containing 10% FCS and antibiotics (100 IU/mL penicillin, 50 μg/mL streptomycin; both from Gibco) and allowed to adhere for three hours. GMSCs were afterwards treated with mitomycin C (Bristol Caribbean Inc., Mayaguez, PR, USA) (25 µg/ mL) for 30 min to block their proliferation, and then washed 4 times with 2% FCS/PBS. PBMCs were obtained by gradient centrifugation of peripheral blood of healthy blood donors upon written consent. PBMCs, either unstimulated or stimulated to proliferate with 10 µg/mL of phytohemagglutinin (PHA; Sigma-Aldrich, Darmstadt, Germany), were added to the co-culture at a density of 1 × 10^5^/well. Control cultures were unstimulated or PHA-stimulated PBMCs. Cells were cultured for 3 days and pulsed with 1 µCi/well [^3^H] thymidine (6.7 Ci/mmol; Amersham Biosciences, Amersham, Bucks, UK) for the last 18 h, followed by cell harvesting and radioactivity counting (LS5000TB scintillation counter, Beckman Coulter, Brea, CA, USA). The incorporation of [^3^H] thymidine into PBMCs was expressed as count/minute (CPM). Proliferation was presented as the proliferation index determined by dividing CPM in PHA-stimulated PBMC culture with CPM in control non-stimulated PBMC cultures. Four different H-GMSCs and four different P-GMSC lines with different PBMCs were used, and each experiment was carried out in triplicates. In co-cultures with *P. gingivalis* treated-GMSCs, we used the same protocol as described above for stimulation of GMSCs with *P. gingivalis*. The incorporated radioactivity into control mitomycin C-treated GMSCs was negligibly small (less than 500 CPM).

### 4.12. ELISA Assay

Concentrations of IL-6 and IL-8 in supernatants of *P. gingivalis*-treated GMSCs were determined using the commercial ELISA kits (R&D Systems GmbH, Wiesbaden, Germany) as described in the manufacturer’s protocol. Cytokine levels were determined based on standard curves constructed according to known concentrations of these cytokines.

### 4.13. Statistics

Kruskal-Wallis or Mann-Whitney tests were used for statistical analysis to assess differences between H and P groups or between experimental and appropriate control samples. Differences in mRNA expression between untreated and *P. gingivalis*-treated GMSCs were evaluated using a Ratio paired t-test or Wilcoxon test. Values at *p* < 0.05 or less were considered to be statistically significant. The statistical analysis and graphs were done in GraphPad Prism version 8.0.0 (GraphPad Software, San Diego, CA, USA).

## Figures and Tables

**Figure 1 ijms-23-03510-f001:**
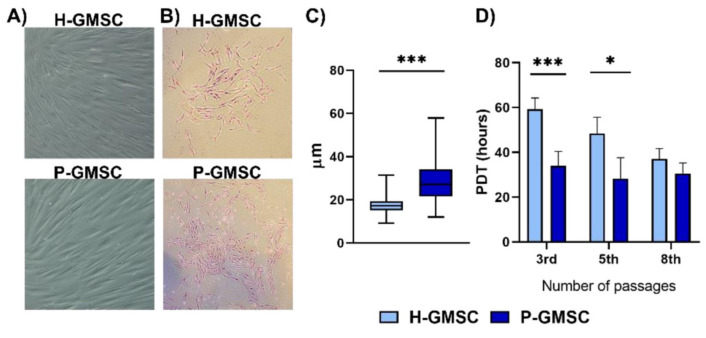
Morphology (**A**), CFU-F (**B**), cell-size (**C**) and population doubling time (PDT) (**D**) of H- and P-GMSC lines. (**A**) and (**B**) are representative images of one of each line, of the three, analysed. (**C**) represents mean ± SD of three lines (each measurement of 50 cells); *** *p* < 0.001. (**D**) represents mean PDT ± SD of three lines in each group; * *p* < 0.05; *** *p* < 0.001.

**Figure 2 ijms-23-03510-f002:**
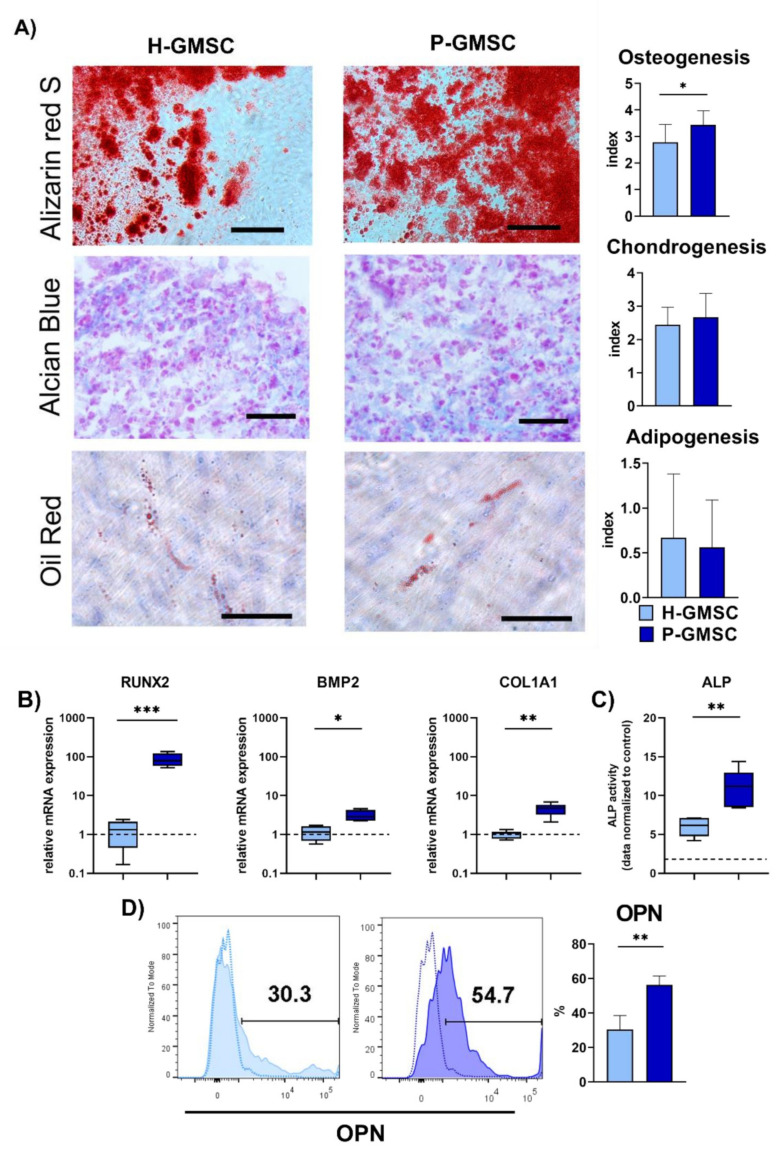
(**A**) Osteogenic (Alizarin red S staining), Chondrogenic (Alcian Blue staining) and Adipogenic (Oil Red staining) of H- and P-GMSC lines. Representative images of one (out of four) differentiation cultures are presented. Vertical columns represent differentiation indices (mean ± SD; *n* = 4). (**B**) RUNX2, BMP-2 and COL1A1 gene expression in H- and P-GMSC lines (*n* = 3) differentiated in the osteogenic medium; (**C**) ALP activity in H- and P-GMSC lines (*n* = 3) differentiated in the osteogenic medium; (**D**) Flow cytometric analysis of OPN expression in H- and P-GMSC lines differentiated in the osteogenic medium. In addition to the total values given for each group (mean ± SD; *n* = 3), representative histograms from one of each line are presented. The bars are placed according to the controls. *** *p* < 0.001; ** *p* < 0.01; * *p* < 0.05 between H- and P-GMSC lines.

**Figure 5 ijms-23-03510-f005:**
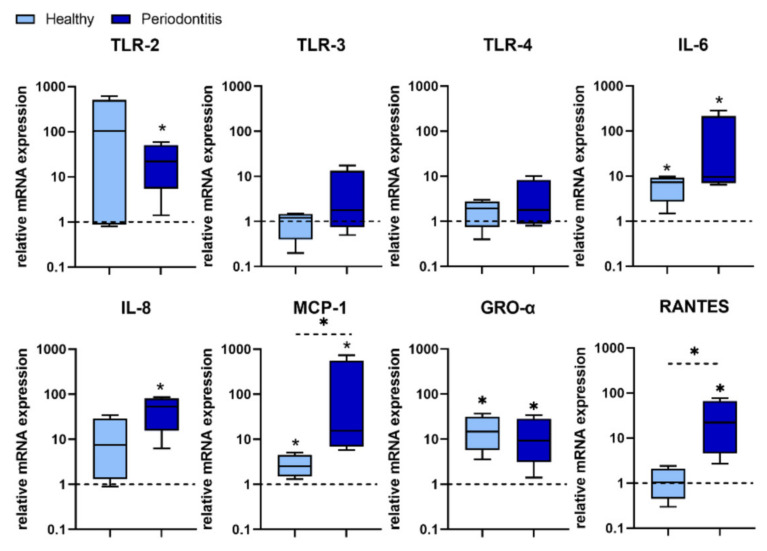
Expression of genes associated with inflammation in GMSCs treated with *P. gingivalis*. The basic level of each mRNA expression is counted as 1 (non-treated GMSCs) and is marked with a dotted line. The asterix above the boxes indicates the significant difference between non-treated and *P. gingivalis* treated H- or P-GMSC lines (* *p* < 0.05). The asterix above the line indicates the significant difference between H- and P-GMSCs treated with *P. gingivalis* (* *p* < 0.05); *n* = 4 in each group.

**Figure 6 ijms-23-03510-f006:**
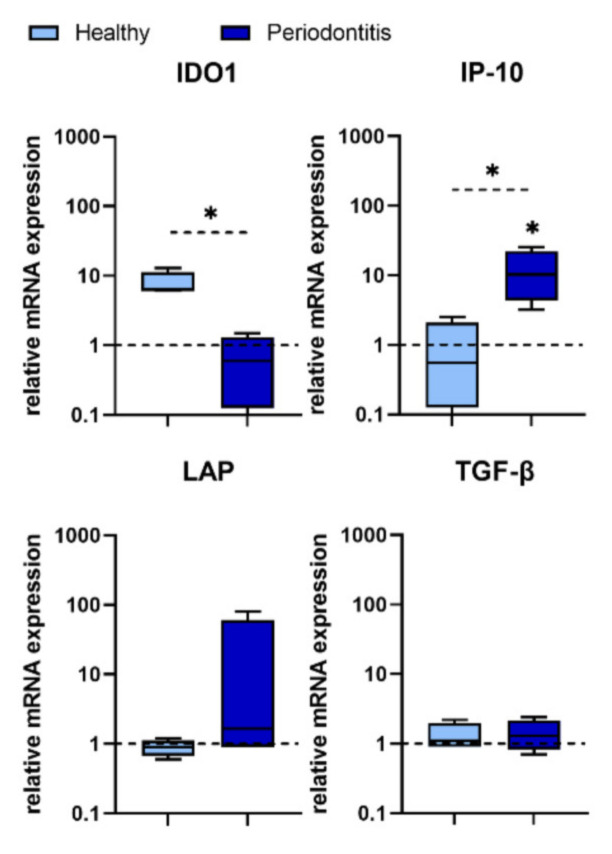
Expression of genes associated with immunomodulation in GMSCs treated with *P. gingivalis*. The basic level of each mRNA expression is counted as 1 (non-treated GMSCs) and is marked with a dotted line. The asterix above the boxes indicates the significant difference between non-treated and *P. gingivalis* treated H- or P-GMSC lines (* *p* < 0.05). The asterix above the line indicates the significant difference between H- and P-GMSCs treated with *P. gingivalis* (* *p* < 0.05); *n* = 4 in each group.

**Figure 7 ijms-23-03510-f007:**
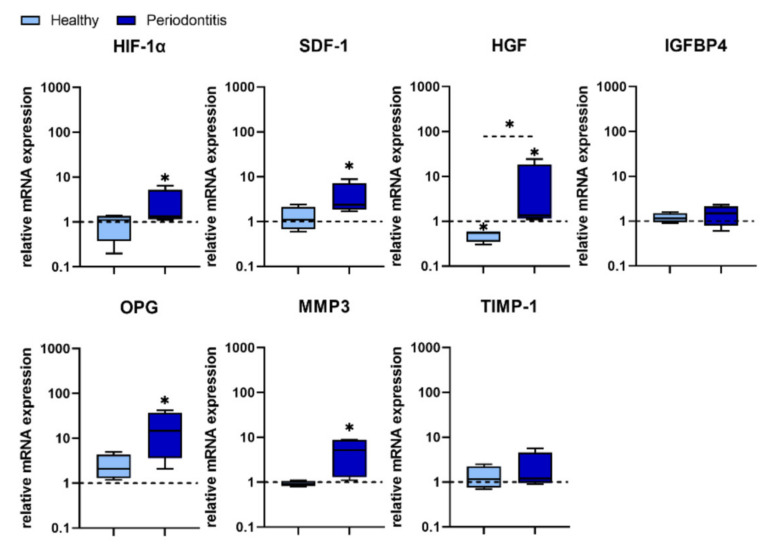
Expression of genes associated with tissue regeneration/repair in GMSCs treated with *P. gingivalis.* The basic level of each mRNA expression is counted as 1 (non-treated GMSCs) and is marked with a dotted line. The asterix above the boxes indicates the significant difference between non-treated and *P. gingivalis* treated H- or P-GMSC lines (* *p* < 0.05). The asterix above the line indicates the significant difference between H- and P-GMSCs treated with *P. gingivalis* (* *p* < 0.05); *n* = 4 in each group.

**Table 1 ijms-23-03510-t001:** Sequences of the primer pairs used for the real-time PCR experiments.

Primers	Sequence
*h IP-10* forward *IP-10* reverse	AGCAGAGGAACCTCCAGTCT ATGCAGGTACAGCGTACAGT
*HGF* forward *HGF* reverse	GCACTGACTCCGAACAGGAT CAGGAGTTTGGTCACCCACA
*MCP-1* forward *MCP-1* reverse	GATCTCAGTGCAGAGGCTCG TTTGCTTGTCCAGGTGGTCC
*HIF-1α* forward *HIF-1α* reverse	GTCTGAGGGGACAGGAGGAT CTCCTCAGGTGGCTTGTCAG
*IGFBP-4* forward *IGFBP-4* reverse	TCTGAGCCCTGGTGTGTTTC GCTGGCACGTAGTACATGGT
*GRO-α* forward *GRO-α* reverse	CTGGCTTAGAACAAAGGGGCT TAAAGGTAGCCCTTGTTTCCCC
*LAP* forward *LAP* reverse	ACTGCCCAGTTCAAGAGACG CCGACCGGATCTGTACTTCG
*RANTES* forward *RANTES* reverse	CAGTCGTCTTTGTCACCCGA CGGGTGGGGTAGGATAGTGA
*OPG* forward *OPG* reverse	TAACGTGATGAGCGTACGGG GCAGCACAGCAACTTGTTCA
*SDF-1* forward *SDF-1* reverse	GGACTTTCCGCTAGACCCAC GTCCTCATGGTTAAGGCCCC
*IDO-1* forward *IDO-1* reverse	GGGAAGCTTATGACGCCTGT CTGGCTTGCAGGAATCAGGA
*TLR-3* forward *TLR-3* reverse	CCTTTTGCCCTTTGGGATGC TGAAGTTGGCGGCTGGTAAT
*TLR-2* forward *TLR-2* reverse	TGAGCTGCCCTTGCAGATAC TGCAAGCAGGATCCAAAGGA
*TLR-4* forward *TLR-4* reverse	GGATTTCACACCTCCACGCA GGTCAGAGCGTGATAGCGAG
*MMP-3* forward *MMP-3* reverse	TGAAATTGGCCACTCCCTGG GGAACCGAGTCAGGTCTGTG
*TIMP-1* forward *TIMP-1* reverse	TCGTCATCAGGGCCAAGTTC TCCACAAGCAATGAGTGCCA
*TGF-β* forward *TGF-β* reverse	CCGGGTTATGCTGGTTGTACAG AAGGACCTCGGCTGGAAGTGG
*IL-6* forward *IL-6* reverse	CACTCACCTCTTCAGAACGA CTGTTCTGGAGGTACTCTAGG
*IL-8* forward *IL-8* reverse	ACACAGAGCTGCAGAAATCAGG GGCACAAACTTTCAGAGACAG
*RUNX2* forward *RUNX2* reverse	GCGGTGCAAACTTTCTCCAG TCACTGTGCTGAAGAGGCTG
*BMP-2* forward *BMP-2* reverse	GGGGTGGGGGAAAGGTAATG TCGGGTTATCCAGGTTTTGCT
*COL1A1* forward *COL1A1* reverse	TCGGAGGAGAGTCAGGAAGG AACAGAACAGTCTCTCCCGC
*OCN* forward *OCN* reverse	GACTGTGACGAGTTGGCTGA CACATCCATAGGGCTGGGAG
*OPN* forward *OPN* reverse	CATACAAGGCCATCCCCGTT GGGTTTCAGCACTCTGGTCA
*β-actin* forward *β-actin* reverse	TCAGTAACAGTCCGCCTAGAAGCA TTGCTGACAGGATGCAGAAGGAGA

## Data Availability

All data are included in this article.

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
