# Peer review of "Mesenchymal Stromal Cells from Healthy and Inflamed Human Gingiva Respond Differently to Porphyromonas gingivalis"

_ijms, 2022, doi:10.3390/ijms23073510_

Round 1
Reviewer 1 Report
The authors investigated the response of GMSCs from healthy and periodontitis patients to the treatment of Porphyromonas gingivalis. The results demonstrated that P-GMSCs have higher proliferative and osteogenic potential than H-GMSCs. Their responses to P. gingivalis treatment differed in terms of inflammatory properties, immunosuppressive activity, and regeneration-related genes.
The objective of the study is clear, the experiments are well organized and conducted, and the data shown support their conclusion. However, there are a few concerns.
- What is the general condition of these patients? The condition of the gingival tissue can also be affected by aging or by systemic diseases.
- In Figure 2, P-GMSC have high osteogenic potential than H-GMSC. Are there any changes in osteogenesis-related genes or protein markers that further support this conclusion?
- The results show that after treatment with P. gingivalis, P-GMSCs have higher pro-inflammatory properties than H-GMSCs, but lower immunosuppressive ability, how could this be explained?
Author Response
Reviewer: The authors investigated the response of GMSCs from healthy and periodontitis patients to the treatment of Porphyromonas gingivalis. The results demonstrated that P-GMSCs have higher proliferative and osteogenic potential than H-GMSCs. Their responses to P. gingivalis treatment differed in terms of inflammatory properties, immunosuppressive activity, and regeneration-related genes.
The objective of the study is clear, the experiments are well organized and conducted, and the data shown support their conclusion. However, there are a few concerns.
1. What is the general condition of these patients? The condition of the gingival tissue can also be affected by aging or by systemic diseases.
Reply
The general status of patients included in this study was added in Materials and methods (page 15, lines 501-505). “Apart from matching by age and sex, subjects with healthy gingiva and patients with periodontitis had no diabetes, systemic autoimmune diseases, or malignancies and were not smokers. In addition, they did not receive antibiotics or immunosuppressive drugs for one month before tissue sampling. Other data from the medical history of study participants were not recorded.”
2. In Figure 2, P-GMSC have high osteogenic potential than H-GMSC. Are there any changes in osteogenesis-related genes or protein markers that further support this conclusion?
Reply
We performed additional experiments to confirm higher osteogenic potential of P-GMSC compared to H-GMSC. It included the study of major osteogenic gene expression (RUNX2, BMP-2, COL1A1), the expression of osteopontin at the protein level and Alkaline phosphatase activity. The results are presented in the new Figure 2B,C and D and are given textually (page 3, lines 133-138). Accordingly, the methodology for AP activity is described in the section 4.7 (page 18, lines 633-648), the sequences of new primers are added in Table 1 (page 20). Antibodies used for flow cytometry are added in the section 4.6 (page 18, lines 622-624).
3. The results show that after treatment with P. gingivalis, P-GMSCs have higher pro-inflammatory properties than H-GMSCs, but lower immunosuppressive ability, how could this be explained?
Reply
We have already provided sufficient data in Discussion that P-GMSC, by their functional characteristics, resemble MSC1 type of MSCs which have pro-inflammatory properties. We now cited an additional reference (new reference 45) to show similar immunosuppressive phenomenon of periodontal ligament stem cells established from periodontitis to emphasize that reduced immunosuppression is needed for promotion of inflammation as a general immunological phenomenon. See the inserted text (page 12, lines 343-349) “In addition, P-GMSCs may have reduced immunosuppressive potential in PBMC cultures, which is further potentiated by P. gingivalis, due to their lower ability to produce IL-10 and induce differentiation of T regulatory cells (Tregs). Such a result was published for periodontal ligament derived MSCs isolated from periodontitis using the same co-culture model, suggesting that a reduction in immunosuppressive mechanisms is needed to promote acute inflammation [45].” Accordingly, the last sentence of the abstract (lines 30-31) is slightly modified. In addition, some errors in English were corrected.
Reviewer 2 Report
Dear Authors,
I want to congratulate you for your hard work. After the analysis of ”Mesenchymal Stromal Cells from Healthy and Inflamed Human Gingiva Respond Differently to Porphyromonas gingivalis” manuscript, I consider it suitable for publication, after addressing the following minor details:
- You need to English-proof your manuscript (UK and US English spelling are mixed in your writing).
- There is a spelling error in Line 560 – ”Barcelona”.
Author Response
Reviewer: I want to congratulate you for your hard work. After the analysis of ”Mesenchymal Stromal Cells from Healthy and Inflamed Human Gingiva Respond Differently to Porphyromonas gingivalis” manuscript, I consider it suitable for publication, after addressing the following minor details:
Reply
Thank you very much for the objectively presented quality of our work.
You need to English-proof your manuscript (UK and US English spelling are mixed in your writing).
Reply
Corrected as best as possible
There is a spelling error in Line 560 – ”Barcelona”.
Reply
Corrected
Round 2
Reviewer 1 Report
The authors made sufficient revisions according to the modification comments and clarified the concerns raised by this reviewer. All the changes made to answer the reviewers help to improve the paper. And I suggest that this paper be accepted.